# Serum Selenium Level and 10-Year Survival after Melanoma

**DOI:** 10.3390/biomedicines9080991

**Published:** 2021-08-11

**Authors:** Emilia Rogoża-Janiszewska, Karolina Malińska, Piotr Baszuk, Wojciech Marciniak, Róża Derkacz, Marcin Lener, Anna Jakubowska, Cezary Cybulski, Tomasz Huzarski, Bartłomiej Masojć, Jacek Gronwald, Helena Rudnicka, Andrzej Kram, Magdalena Kiedrowicz, Magdalena Boer, Tadeusz Dębniak, Jan Lubiński

**Affiliations:** 1Department of Genetics and Pathology, International Hereditary Cancer Center, Pomeranian Medical University, 71-252 Szczecin, Poland; karolina.malinska@pum.edu.pl (K.M.); baszukpiotr@gmail.com (P.B.); wojciech.marciniak90@gmail.com (W.M.); roza.derkacz@gmail.com (R.D.); marcinlener@poczta.onet.pl (M.L.); aniaj@pum.edu.pl (A.J.); cezarycy@pum.edu.pl (C.C.); huzarski@pum.edu.pl (T.H.); jgron@pum.edu.pl (J.G.); helena.rudnicka@pum.edu.pl (H.R.); debniak@pum.edu.pl (T.D.); lubinski@pum.edu.pl (J.L.); 2Radiation Oncology Department, West Pomeranian Oncology Center, 71-730 Szczecin, Poland; bmasojc@gmail.com; 3Pathology Department, West Pomeranian Oncology Center, 71-730 Szczecin, Poland; akram@onkologia.szczecin.pl; 4Department of Skin Diseases and Venerology, Pomeranian Medical University in Szczecin, 72-010 Police, Poland; magkied@gmail.com (M.K.); m.boer@onet.eu (M.B.)

**Keywords:** melanoma, selenium, survival

## Abstract

Melanoma is one of the most aggressive human malignancies. The determination of prognostic biomarkers is important for the early detection of recurrence and for the enrollment of the patients into different treatment regimens. Herein, we report the 10-year survival of 375 melanoma patients depending on their serum selenium levels. The study group was followed up from the date of melanoma diagnosis until death or 2020. Patients were assigned to one of four categories, in accordance with the increasing selenium level (I–IV quartiles). The subgroup with low selenium levels had a significant lower survival rate in relation to patients with high selenium levels, HR = 8.42; *p* = 0.005 and HR = 5.83; *p* = 0.02, for uni- and multivariable models, respectively. In the univariable analysis, we also confirmed the association between Breslow thickness, Clark classification and age at melanoma prognosis. In conclusion, a low serum selenium level was associated with an increased mortality rate in the 10 years following melanoma diagnosis. Future studies in other geographic regions with low soil selenium levels should be conducted to confirm our findings.

## 1. Introduction

Melanoma is one of the most aggressive human malignancies. Over 320,000 new cases were diagnosed globally in 2020, with 57,000 deaths. Data from the Global Cancer Observatory indicate that over the last decade, the incidence rates of melanoma have increased by nearly 50%, with deaths increasing by 32%. According to the WHO prediction, the number of deaths related to melanoma will increase by 20% in 2025, rising to 74% in 2040 [1]. The American Cancer Society, based on information from the SEER database, maintained by the National Cancer Institute (NCI), estimates the 5-year survival rate to be 99% in localized, 66% in regional and 23% in distant melanoma SEER stages. Although the TNM (Tumor Node and Metastasis) staging, Clark classification and Breslow thickness are the most frequently used clinical parameters for melanoma prognosis, a precise prognosis of survival within the same staging groups requires more parameters. The determination of prognostic biomarkers is important for the early detection of recurrence and for the enrollment of the patients into different treatment regimens [2]. Selenium is an essential component of several major metabolic pathways, including the antioxidant defense system and the immune system, and selenium is incorporated into 30 different selenoproteins [3,4,5]. Selenium has an effect on cell proliferation and apoptotic cell death in healthy and malignant cells [6]. Low selenium concentrations have been associated with a high incidence of several different cancer types [5,7] as well as cancer mortality [5,8,9]. Too high a selenium concentration (selenosis) is also correlated with a higher occurrence of some common diseases [7]. A close correlation between serum selenium (Se) concentrations and the risk of death, regardless of the cause, has been reported [4]. The mortality patterns related to long-term exposure to inorganic hexavalent selenium through drinking water were elevated for melanoma disease [10].

To date, there have been no studies aimed at evaluating the role of the serum selenium concentration and melanoma prognosis. Herein, we report the 10-year survival of our cohort of melanoma patients depending on their serum selenium levels.

## 2. Materials and Methods

### 2.1. Study Participants

A total of 375 melanoma patients were enrolled into the study after providing written informed consent. They were selected from a registry of 1500 malignant melanoma (MM) cases with histopathologically confirmed disease, housed at the Hereditary Cancer Center in Szczecin and diagnosed between 2006 and 2016 in Polish cities (Szczecin, Gorzow Wielkopolski, Opole, Bialystok, Zielona Gora). All newly diagnosed melanoma cases with a secured biobank were enrolled in the study. This study was conducted in accordance with the Declaration of Helsinki, and all participants signed a written informed consent document prior to donating a blood sample for analysis. This study was approved by the Ethics Committee of the Pomeranian Medical University in Szczecin (number KB-0012/73/10). All patient blood samples were collected at the time of melanoma diagnosis, but before the commencement of any treatment other than surgical removal of skin lesions. Consenting patients were asked to fast for at least four hours prior to blood collection. A blood sample (10 cc) was obtained during the diagnostic workup and was collected into tubes certified for quantification of trace metals (Vacutainer^®^ System, Becton Dickinson, USA, royal blue cap). Blood samples were taken between 8 a.m. and 2 p.m. and were centrifuged within 30 to 120 mins of collection to separate the serum from the cellular fraction. The serum samples were stored at −80 °C until being required for the selenium assay.

### 2.2. Measurement of Selenium Level

Serum selenium levels were quantified by inductively coupled mass spectrometry (ICP-MS NexION 350D, Perkin Elmer) using methane for reduction in polyatomic interferences. Calibration standards were prepared by dilution of 10 mg/l Multi-Element Calibration Standard 3 (PerkinElmer Pure Plus, PerkinElmer Life and Analytical Sciences, USA) with reagent blank consisting of 0.65% solution of nitric acid (Suprapur, Merck, Germany) and 0.002% Triton X-100 (PerkinElmer, Tokyo, Japan, USA). Calibration curves were created using four different concentrations: 0.1 μg/L, 0.5 μg/L, 1 μg/L, 2 μg/L. Germanium (PerkinElmer Pure, PerkinElmer Life and Analytical Sciences, USA) was used as an internal standard, and ClinChek^®^ Plasma Control Level I (Recipe, Munich, Germany) was used as a reference material. Reference material was measured after each of the six samples. If the difference in the reference material measurements was greater than 5%, the entire series was repeated. Each sample was measured in duplicate from different analytical runs. Prior to analysis, all samples were centrifuged (4000 g, 15 min), and the supernatant was diluted 100 times with the reagent blank. Technical details, plasma operating settings and mass spectrometer acquisition parameters are available on request.

### 2.3. Statistical Analysis

A total of 375 melanoma patients were enrolled into the study after providing written informed consent. Patients were assigned to one of four categories, equal in terms of the number of participants, in accordance with the increasing selenium level (I quartile (56.68–76.23), II quartile (76.44–85.01), III quartile (85.15–96.06), IV quartile (96.15–168.01)).

The study group was followed up from the date of melanoma diagnosis until death or 2020. The range of follow-ups varied between 0 and 10 years (observation time longer than 10 years is considered as 10-year observation). Univariable and multivariable Cox proportional hazard models were used in order to calculate the hazard ratios (HR). The multivariable analysis took into account the following factors: selenium levels (divided into quartiles), Clark (II—invasion of the basal layer epidermis, III—invasion of the papillary dermis, IV—invasion of the reticular dermis, V—invasion of the subcutaneous fat), sex (male/female) and age (continuous variable). Due to relatively small number of cases with Clark V, we decided to include them into the Clark IV category for calculation purposes. Subjects with Clark I (melanoma in situ) were excluded from the database according to the fact that this type of melanoma does not have any influence on mortality. Due to missing data according to the Breslow variable, additional calculations were performed on subgroups containing complete data according to the mentioned factor (*n* = 324). The graphical representation of the survival rates in the univariable approach is covered by Kaplan–Meier curves. Patients with the highest selenium levels (IV quartile) were chosen as a reference due to having the highest alive/deceased ratio. In order to calculate differences between selenium levels between the sexes and Clark factors, an unpaired Wilcoxon rank sum test was applied (due to data distribution in compared groups being different than normal). All calculations were performed using: “R: A language and environment for statistical computing. R Foundation for Statistical Computing, Vienna, Austria” (R version 4.0.4 (2020–10–10).

## 3. Results

The characteristics of the patients are presented in Table 1. The mean age at diagnosis was 54.63 years (range 21–90 years). The majority of the patients were females (62%), and 39% were diagnosed with Clark IV/V disease. The overall 10-year survival was 40.1% for the entire cohort. The Kaplan–Meier survival curves in relation to the quartiles of serum selenium levels are presented in Figure 1. The median Se level among all malignant melanoma (MM) patients was 85.15 μg/L, the interquartile range (IQR) was 19.77 μg/L and the mean Se level was 87.91 μg/L (range from 56.68 to 168.01 μg/L). 

No significant difference in the selenium concentration was observed in relation to sex (*p* = 0.7362). 

The median Se level was lower (*p* = 0.04987) among Clark IV/V cases (median 82.16 μg/L) when compared to Clark III (median 84.51 μg/L) or Clark II (median 89.9993 μg/L) patients (*p* = 0.001235). The mean, standard deviation, median, range and IQR are shown in Table 2. The hazard ratios, confidence intervals and *p*-values for the uni- and multivariable Cox proportional hazard regression models for all analyzed factors are presented in Table 3 and Table 4. The beneficial effect of the selenium concentration on survival was observed in patients with the highest selenium concentrations. The subgroup with low selenium levels (I quartile) had a significant lower survival rate in relation to patients with high selenium levels (IV quartile), HR = 8.42; *p* = 0.005 and HR = 5.83; *p* = 0.02, for uni- and multivariable models, respectively. Men had a higher HR than women; however, in the univariable model, the result is on the border of significance (HR 2.12; *p* = 0.037). In the multivariable models, the differences between HRs depending on sex were not significant (HR = 1.58 *p* = 0.2). Hazard ratios were significantly higher with increasing age (HR= 1.06; *p* < 0.001 and HR = 1.05; *p* = 0.002, for uni- and multivariable models, respectively). Melanoma patients with Clark IV/V had higher hazard ratios compared to the patients with Clark II (HR = 8.76; *p* = 0.035) in the univariable model, but this was not significant in the multivariable model (HR = 5.11; *p* = 0.12).

In additional analysis, in the group with complete data according to the Breslow thickness (*n* = 324), a difference in the survival rate depending on the Se level can be also observed; however, due to the smaller sample size, the difference between patients with the lowest Se levels (I quartile) compared to those with the highest levels (IV quartile) is on the border of significance in the multivariable model (HR = 7.26; *p* = 0.062). Significance in relation to Se levels can be still observed in the univariable modeling (HR 10.6; *p* = 0.025). The detailed uni- and multivariable results for the subgroup taking into account the Breslow factor as an independent variable are presented in Table 4. 

Additionally, in this subgroup, the age factor seems to have a significant relation with survival among MM patients (HR = 1.06; *p* = 0.002 and HR = 1.05; *p* = 0.017, for uni- and multivariable regression models).

^1^ HR = hazard ratio, CI = confidence interval.

## 4. Discussion

The literature data are inconsistent regarding the possible association of selenium and melanoma. In a murine study, a dose-dependent difference was found with selenium and melanoma development, with the moderate dosage increasing tumor growth, and the high dosage effectively treating and preventing the recurrence of fully malignant tumors [11].

In several studies, no relation was reported between either toenail selenium concentrations or self-reported selenium supplement use and melanoma risk [12,13,14,15]. 

On the other hand, an increased concentration of plasma selenium was found to be associated with an increased risk of melanoma among an Italian population. In the same study, toenail and dietary selenium exhibited no evidence of a relation with melanoma risk; this difference could have been, in part, due to differences in specific selenium compounds [16]. A higher incidence of melanoma has been reported among individuals exposed to unusually high levels of inorganic hexavalent selenium (selenate) through drinking water [17].

The authors of a recent meta-analysis of the literature data concluded that studies demonstrate the need to further investigate the exposure classification of selenium biomarkers, and metabolism of selenium, in order to elucidate the potential relation between selenium exposure and melanoma risk [18].

To our knowledge, melanoma prognosis according to the selenium concentration has not been studied up to now. Herein, in the study of 375 melanoma patients from Poland, we found that a low serum selenium level (i.e., below 76.44 μg/L) was associated with an increased mortality rate in the 10 years following diagnosis. All study participants fasted before blood sample collection for selenium level evaluation. The measurement was conducted prior to treatment other than surgical removal of the primary skin lesion. Additionally, none of the host factors (e.g., Clark, Breslow status) were associated with selenium concentrations, and it is likely that the association is due to unrecognized confounding. Our study has several limitations. We had no data on BMI status. Selenium was measured only once, and a single serum measurement reflects short-term selenium intake. Although the patient cohort was relatively large, the small sample sizes for various subgroups were not well powered in our subgroup analyses. However, we saw a significant association between selenium and melanoma survival in both the univariable and multivariable analyses. The association was restricted to patients with a low selenium level, and a trend in survival across the four quartiles was observed. In the univariable analysis, we also confirmed the association between Breslow thickness, Clark classification and the age at melanoma prognosis. 

Selenium takes part in several cellular processes and molecular pathways that may be involved in anti-cancer activity, i.e., reduction in DNA damage, oxidative stress and inflammation; detoxification of carcinogens; enhancement of immune response; alteration in DNA methylation; regulation of the cell cycle; induction of apoptosis of cancer cells; inhibition of angiogenesis required for the growth and metastasis of tumors [5,19,20,21,22,23,24,25,26,27,28]. Additionally, many other in vivo and in vitro reports have found that selenium compounds and selenoproteins may affect cell motility, migration and invasion [20,22,29,30].

Although it is not definitely understood how selenium affects cancer prognosis, the literature data support the thesis of an association between selenium levels and cancer outcome. Harris et al. suggested that selenium may improve breast cancer-specific survival and overall survival [31]. 

Other publications have recently presented the association between low levels of circulating selenium and survival in women with breast cancer [32,33,34]. 

Higher serum selenium levels at the time of diagnosis have also been reported to be associated with improved outcome in patients with cancers of the lung and larynx [35,36].

Our findings are consistent with the literature data, since in all published studies, either low serum selenium levels correlated with worse prognosis or higher selenium levels correlated with a better outcome. 

## 5. Conclusions

We showed, herein, that a low selenium level might contribute to worse survival for patients with melanoma. Future studies in other geographic regions with low soil selenium levels should be conducted to confirm our findings.

## Figures and Tables

**Figure 1 biomedicines-09-00991-f001:**
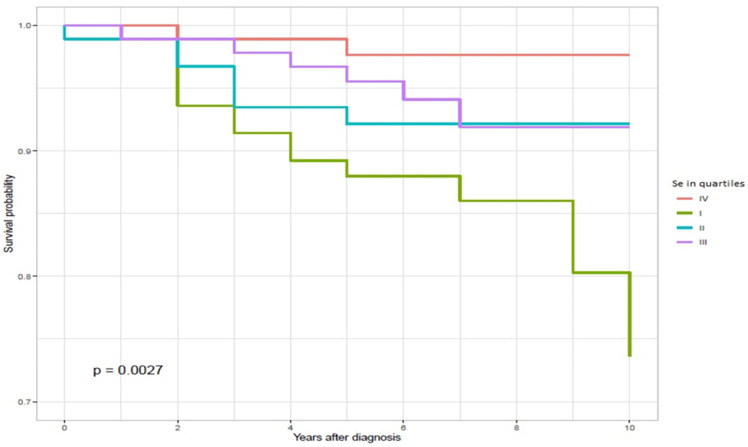
Ten-year survival by serum selenium levels (quartiles) in MM patients.

**Table 1 biomedicines-09-00991-t001:** Characteristics of the study group (*n* = 375).

Characteristic	Overall, *n* = 375 ^1^	Alive, *n* = 344 ^1^	Deceased, *n* = 31 ^1^
Selenium level in quartiles (µg/L)			
I (56.68–76.23)	94 (25%)	78 (23%)	16 (52%)
II (76.44–85.01)	93 (25%)	86 (25%)	7 (23%)
III (85.15–96.06)	94 (25%)	88 (26%)	6 (19%)
IV (96.15–168.01)	94 (25%)	92 (27%)	2 (6.5%)
Sex			
Female	232 (62%)	218 (63%)	14 (45%)
Male	143 (38%)	126 (37%)	17 (55%)
Age	21.00–90.00 (54.63)	21.00–90.00 (53.76)	38.00–86.00 (64.26)
Breslow (mm) *	0.20–16.80 (1.80)	0.20–16.80 (1.71)	0.50–11.00 (3.22)
Clark II	71 (19%)	70 (20%)	1 (3.2%)
Clark III	157 (42%)	145 (42%)	12 (39%)
Clark IV/V	147 (39%)	129 (38%)	18 (58%)

^1^ n (%); range (mean), * *n* = 324.

**Table 2 biomedicines-09-00991-t002:** Selenium levels by subgroups (*n* = 375).

Subgroup	Level	*n*	Mean	SD	Median	Min	Max	Range	IQR
Sex									
	Female	232	88.29	18.15	85.25	56.68	168.01	111.33	19.48
	Male	143	87.29	16.83	84.46	58.17	162.96	104.79	19.59
Clark									
	II	71	92.77	19.35	89.99	62.61	168.01	105.4	16.93
	III	157	88.78	18.43	84.51	57.72	166.08	108.36	21.78
	IV/V	147	84.63	15.22	82.16	56.68	143.66	86.98	19.82
Selenium level in quartiles (µg/L)									
	I	94	69.99	5.2	71.94	56.68	76.23	19.55	6.47
	II	93	80.62	2.69	80.96	76.44	85.01	8.56	5.08
	III	94	90.29	3.31	90.19	85.15	96.06	10.91	6.42
	IV	94	110.66	17.3	102.57	96.15	168.01	71.87	17.24

**Table 3 biomedicines-09-00991-t003:** Univariable and multivariable Cox proportional hazard models for given factors for 375 melanoma patients.

	Univariable Cox Regression			Multivariable Cox Regression		
Characteristic	HR ^1^	95% CI ^1^	*p*-Value	HR ^1^	95% CI ^1^	*p*-Value
Selenium level in quartiles (µg/L)						
I (56.68–76.23)	8.42	1.94, 36.6	0.005	5.83	1.32, 25.8	0.020
II (76.44–85.01)	3.73	0.77, 18.0	0.10	3.37	0.70, 16.3	0.13
III (85.15–96.06)	3.05	0.62, 15.1	0.2	3.34	0.67, 16.7	0.14
IV (96.15–168.01)	—	—		—	—	
Sex						
Female	—	—		—	—	
Male	2.12	1.05, 4.31	0.037	1.58	0.77, 3.27	0.2
Age	1.06	1.03, 1.09	<0.001	1.05	1.02, 1.08	0.002
Breslow (mm) *	1.16	1.04, 1.29	0.008	1.10	0.96, 1.27	0.2
Clark II	—	—		—	—	
Clark III	5.47	0.71, 42.1	0.10	3.94	0.50, 31.0	0.2
Clark IV/V	8.76	1.17, 65.7	0.035	5.11	0.67, 39.3	0.12

^1^ HR = hazard ratio, CI = confidence interval, * *n* = 324.

**Table 4 biomedicines-09-00991-t004:** Univariable and multivariable Cox proportional hazard models for given factors for 324 melanoma patients.

	Univariable Cox Regression			Multivariable Cox Regression		
Characteristic	HR^1^	95% CI ^1^	*p*-Value	HR ^1^	95% CI ^1^	*p*-Value
Selenium level in quartiles (µg/L)						
I (56.68-76.23)	10.6	1.35, 82.5	0.025	7.26	0.90, 58.3	0.062
II (76.44–85.01)	5.33	0.62, 45.6	0.13	4.92	0.57, 42.7	0.15
III (85.15- 96.06)	4.17	0.47, 37.3	0.2	4.97	0.54, 45.4	0.2
IV (96.15–168.01)	—	—		—	—	
Sex						
Female	—	—		—	—	
Male	1.55	0.64, 3.75	0.3	1.16	0.47, 2.87	0.7
Age	1.06	1.02, 1.10	0.002	1.05	1.01, 1.09	0.017
Breslow (mm)	1.16	1.04, 1.29	0.008	1.10	0.96, 1.27	0.2
Clark						
II	—	—		—	—	
III	3.90	0.49, 30.8	0.2	3.22	0.40, 26.1	0.3
IV/V	5.11	0.65, 40.0	0.12	2.44	0.28, 21.2	0.4

^1^ HR = hazard ratio, CI = confidence interval.

## Data Availability

The datasets analyzed during the current study are available from the corresponding author on reasonable request.

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
