# Peer review of "Serum Selenium Level and 10-Year Survival after Melanoma"

_biomedicines, 2021, doi:10.3390/biomedicines9080991_

Round 1
Reviewer 1 Report
The paper investigates an area not previously covered. The findings are original, relevant to practice, novel, may influence the management of melanoma in future and convincing that more future work is needed to verify these findings. Selenium intake modification may well influence melanoma prognosis and aspects of therapy in future. This is very important for a disease that is escalating in prevalence, has global spread and a quite concerning mortality rate. >From both personal and professional perspectives I find the findings very interesting and see this as a potential seminal paper in future. In Australia, melanoma is the current leading cause of cancer deaths in young adults (men and women). The references are relevant and appropriate.
Author Response
Reviewer 1:
The paper investigates an area not previously covered. The findings are original, relevant to practice, novel, may influence the management of melanoma in future and convincing that more future work is needed to verify these findings. Selenium intake modification may well influence melanoma prognosis and aspects of therapy in future. This is very important for a disease that is escalating in prevalence, has global spread and a quite concerning mortality rate. >From both personal and professional perspectives I find the findings very interesting and see this as a potential seminal paper in future. In Australia, melanoma is the current leading cause of cancer deaths in young adults (men and women). The references are relevant and appropriate.
We are agree with reviewer’s comments.

Reviewer 2 Report
The manuscript analyses melanoma patient
prognosis that is dependent on selenium level in serum.
There are several minor points to address:
1 Please, add a paragraph summarizing
different studies on selenium/dependent
cellular processes, involvement in cancer
biology.
2. It would be nice to include a
Figure with selenium structure and
pathways, signaling partners in Introduction.
3. Please add p-value to all Figure graphs.
4. Format references in same way.
Author Response
Reviewer 2:
The manuscript analyses melanoma patient prognosis that is dependent on selenium level in serum.
There are several minor points to address:
1 Please, add a paragraph summarizing different studies on selenium/dependent cellular processes, involvement in cancer biology.
In the discussion section, we added a paragraph summarizing various studies on selenium and selenium-dependent cellular processes involved in cancer biology.
- It would be nice to include a Figure with selenium structure and pathways, signaling partners in Introduction.
In such a short time we are not able to create own figure or receive permission to publish a figure from another source. If this is necessary to proceed to further stages of submission of our manuscript, we need more time to include this figure in our manuscript.
- Please add p-value to all Figure graphs.
We have added a new Figure 1 with the calculated p-value in our manuscript.
- Format references in same way.
We have corrected bibliography according to the changes made.
